

Investigating the Impact of Saharan Dust Aerosols on Analyses and Forecasts of African
Easterly Waves by Constraining Aerosol Effects in Radiance Data Assimilation

By

Dustin F. P. Grogan[1]

Cheng-Hsuan Lu[1,2]

Shih-Wei Wei[1,2]

Sheng-Po Chen[1,3]

1. University at Albany, State University of New York, Albany, NY
2. Joint Center for Satellite Data Assimilation, Boulder, CO
3. Department of Chemistry, National Central University, Taoyuan, Taiwan

Submitted to

Atmospheric Chemistry and Physics

on

February 15[th], 2021

Revised on

January 25[th], 2022

Corresponding author: Dustin Grogan, University at Albany, 1400 Washington Ave, Albany, NY
12222; dgrogan@albany.edu

**Abstract**

This study incorporates aerosol effects into satellite radiance calculations within the Global Data Assimilation System (GDAS) to investigate its impact on the analyses and forecasts of African easterly waves (AEWs). Analysis fields from the aerosol-aware assimilation experiment were compared to an aerosol-blind control during August 2017. The results showed that the aerosol-aware assimilation warmed the Saharan boundary layer, accelerated the African easterly jet, and modified the time-averaged AEWs by enhancing the northern track and reducing the southern track. The changes to the tracks are qualitatively consistent with arguments of baroclinic and barotropic instability. During the time period, we also examined two AEWs that developed Hurricanes Gert and Harvey over the Atlantic, but were structurally different over Africa; the AEW for Gert consisted of a southern vortex, while the AEW for Harvey consisted of a northern and southern vortex. Analysis differences of the cases showed stronger vorticity changes for the AEW that developed Harvey, which we attribute to the aerosol-aware assimilation capturing the radiative effects of a large-scale Saharan dust plume interacting with the northern vortex of the wave. Subsequent forecasts for the AEW cases using the Global Forecast System (GFS, v14) showed that the aerosol-aware assimilation reduced errors in the downstream vorticity structure for the AEW that developed Harvey; neutral improvement was found for the AEW that develop Gert. Thus, aerosol-affected radiances in the assimilation system have the ability to account for dust radiative effects on the analyzed AEWs, which in turn can improve the forecasting of AEWs downstream.

**1. Introduction**


In regions around the world, aerosols can have a profound impact on weather. This is

especially the case over North Africa as it houses the Saharan Desert, which is the largest emitter
of mineral dust aerosols, and African Easterly Waves (AEWs), which bring crucial rainfall to
populations in the Sahel.

AEWs are the dominant synoptic-scale disturbance over North Africa from March to

October (Carlson 1969; Burpee 1972). The waves develop along the African easterly jet (AEJ),
which is a tropospheric jet (~650 hPa) whose axis is centered in the Sahel (~15°N). The AEWs
are also maintained by the AEJ through barotropic and baroclinic energy conversions (Norquist
et al. 1977).  Consequently, the AEWs can have two cyclonic circulations (vortices) that reside
on either side of the AEJ axis (Reed et al. 1988; Pytharilous and Thorncroft 1999). The vortex
south of the AEJ peaks at ~650 hPa and is frequently coupled to moist convection (Kiladis et al.
2006; Berry and Thorncroft 2005), while the northern vortex peaks at ~850 hPa, is dry, and can
be immersed in Saharan dust (Knippertz and Todd 2010; Grogan and Thorncroft 2019). Over the
East Atlantic, the two vortices often merge into a single vortex, which can produce a favorable
environment for tropical cyclogenesis (Schwendike and Jones 2010; Ross and Krishnamurti

2007).

During summer, Saharan dust emissions are most active over the western Sahel (16°N-

24°N, 0°-15°W) (Cowie et al. 2014), the same region the AEW northern track resides. The
emissions are driven by enhanced surface winds that blow over dry and erodible regions (Tegan
and Fun 1994; Webb and Strong 2011). Once lifted, the dust mixes within the deep Saharan
boundary layer (Cuesta et al. 2009; Knippertz and Todd 2012) and can form plumes that span
thousands of kilometers. The transport of these large-scale dust plumes has been connected to
African easterly waves (Westphal et al. 1988; Jones et al. 2003; Knippertz and Todd 2010;
Nathan et al. 2019; Grogan and Thorncroft 2019; Grogan and Nathan 2021). The dust can also be
carried westward over the Atlantic within the Saharan air layer (SAL) (Karyampudi et al. 1999;
Chen et al. 2010), which is an elevated layer of dry air that originates from the Saharan boundary
layer.

Dust directly affects the scattering and absorption of incoming and outgoing radiation of

the atmosphere, which produces heating rates that can influence AEWs through two distinct
pathways (Bercos-Hickey et al. 2017). The first pathway is through the background (time-
averaged) dust fields, which produce heating rates that modify the background temperature and
wind fields (i.e., the AEJ), which in turn affects AEW structure and development (Jones et al
2004; Wilcox et al. 2010; Jury and Santiago 2010). The second pathway is through the formation
of large-scale episodic dust plumes, which produces heating rates that correlate with the wind
and temperature of the AEW to directly affect its growth rates, phase speeds, energetics, and
spatial structures (Grogan et al. 2016, 2017, 2019; Nathan et al. 2017).

To incorporate the above-mentioned dust radiative effects on AEWs within a numerical

weather prediction (NWP) system, it is important to represent the episodic nature of the aerosols.
These radiative effects have been included into NWP systems through two approaches: (i)
radiatively coupling aerosols in the forecast model, and (ii) incorporating aerosols in satellite
radiance calculations during data assimilation (DA).

For the first approach, aerosol attenuation modifies the heating rates within the radiation

schemes of the forecast model of the NWP system. Studies have shown that this improves the
forecast skill of several features in dust-affected regions over North Africa and the East Atlantic,
including sea-level pressure and atmospheric temperature (Perez et al. 2006; Mulcahy et al.
2014), AEWs linked to tropical cyclogenesis (Reale et al. 2009; Reale et al. 2011; Chen et al.
2015), and the AEJ (Reale et al. 2014). Major efforts are also ongoing to improve aerosol
prediction models, including the particle's emission and removal processes, assimilating
observations such as aerosol optical depth (AOD), and model verification and evaluation (see
Benedetti et al. (2018) for a comprehensive discussion). Such advances in aerosol prediction
models can, in turn, improve weather prediction. But despite these advances, the radiative
coupling of episodic aerosols in the NWP system is often not feasible in an operational setting
due to computational costs. Thus, most operational NWP systems use prescribed aerosol
climatologies, such as the NCEP operational Global Forecast System (GFS; Hou et al. 2002) and
the ECMWF integrated forecast system (IFS; Bozzo et al. 2017). Consequently, the NWP system
sacrifices the ability to represent episodic aerosol signals.

For the second approach, aerosol transmittance effects are considered during radiance

DA, which modifies the analysis fields of the NWP system. Kim et al. (2018) demonstrated this
approach by including 3-hourly aerosol fields from the Goddard Chemistry Aerosol Radiation
and Transport (GOCART) model into the radiance calculations within the Goddard Earth
Observing System (GEOS)-Atmospheric Data Assimilation System (ADAS). Kim et al. (2018)
showed that when aerosols were considered, they found the fit to observations improved for
satellite infrared (IR) sounders due to accounting for the aerosol transmittance effects in the form
of cooling brightness temperatures (BT), which has been observed in previous studies (e.g.,
Sokolik 2002). As a result, the cooling of BTs led to warmer analyzed surface temperatures in
the Tropical Atlantic. Similarly, Wei et al. (2020, 2021) showed that when including aerosols
from NOAA's Environmental Modeling System (NEMS) GFS Aerosol Component (NGAC) into
NCEP's global data assimilation system (GDAS), the aerosol transmittance effects warmed
analyzed sea-surface temperatures and low-level air temperatures over the Atlantic and Africa.
Wei et al. (2020) also showed that the aerosols improved GFS forecasts of vector winds and
geopotential heights at multiple levels in the tropical region.

Incorporating aerosol transmittance effects into the radiance calculation of DA is

excluded from all NWP centers, despite its relatively low computation costs and its potential to
leverage aerosol-affected radiances in a physical and consistent way. But more studies
investigating this approach are needed. For example, no study has used this approach to examine
the impacts of dust radiative effects on AEWs in the NWP system. Motivated by the results in
Kim et. al. (2018) and Wei et al. (2020, 2021), along with the physical understanding of dust
radiative effects on AEWs identified above, this study seeks to examine how, and to what extent,
episodic aerosols in the satellite radiance calculations can affect analyses and forecasts of AEWs
over North Africa and the East Atlantic. We focus our analysis on two AEWs during August
2017 that are structurally different over North Africa but later developed hurricanes over the
Atlantic.

In Section 2, we describe the model experiments and the methods used to track the

AEWs. Section 3 presents the analyses and forecasts from each experiment and examines the
aerosol-aware experiment in the context of dust radiative effects on AEWs. Section 4 provides
conclusions and a short discussion.
**2. Experiments and Methods**
*2.1 Model Experiments*

The schematic in Fig. 1 illustrates the workflow of the experiments in this study, which

were conducted from 25 July – 28 August, 2017. The first experiment is an aerosol blind run
(CTL), where aerosols are not considered in the assimilation system. The second experiment is
an aerosol-aware run (AER), which constrains aerosol transmittance effects into the radiance
calculations of the assimilation system (i.e., aerosol-affected radiances). For our experiments, we
employ version 14 of the National Centers for Environmental Prediction (NCEP) Global
Forecast System (GFS, v14), which consists of an analysis system, the Global Data Assimilation
System (GDAS), and a forecast model, the global spectral model (GSM), with GFS physics. The
experiments are fully-cycled, which means that each analysis is constructed from their respective
forecasts of the prior cycle.
The analyses are constructed using GDAS (Fig. 1: blue), which is a Gridpoint Statistical
Interpolation (GSI) based four-dimensional ensemble-variational (4DEnVar) assimilation
system. The assimilation system is run for 80 ensemble members at T254 (~80km) resolution. In
GDAS, the radiance calculations are conducted by the Community Radiance Transfer Model
(CRTM) (Lu et al. 2021). The CRTM generates simulated brightness temperatures (BT) and
computes the radiance sensitivities with respect to the state variables (Han et al. 2006).
For both experiments, various observations are ingested into GDAS, including the
conventional dataset (e.g., radiosondes, ships, buoys, etc.), and satellite observations (e.g.,
retrievals and radiances) (Fig. 1: gray). For the radiance observations, we include the level 1
product of IR and microwave sensors, which are pre-processed by NOAA's National
Environmental Satellite, Data, and Information Service (NESDIS). For a complete list of the
thermal IR sensors, see Table 1 of Wei et al. (2021).
For AER, aerosol transmittance effects can be constrained in CRTM by ingesting three-
dimensional aerosol mixing ratios into GDAS. CRTM contains look-up tables for aerosol optical
properties—absorption coefficient, single scattering albedo, and asymmetric factor— to compute
the aerosol-affected radiances (Lu et al. 2021). The optical properties are based on the Optical
Properties of Atmospheric Composition (OPAC) software package (Hess et al. 1998).

The aerosol mixing ratios are provided by the NEMS GFS Aerosol Component model

(NGAC, v2) (Fig. 1: gold), which is based on GOCART (Colarco et al. 2010). NGAC simulates
the emission, mixing, transport and removal (wet and dry) for 15 externally mixed aerosols,
including dust, sea salt, sulfate, organic carbon, and black carbon. (Lu et al. 2016; Wang et al.,
2018). The NGAC forecasts are used to predict the aerosol mixing ratios during the analysis
window of each cycle. Like the meteorological fields, the aerosol mixing ratios are interpolated
to the observations in space and time using the First Guess at Appropriate Time (FGAT) (Lorenc
and Rawlins 2005). Figure 2 shows the NGAC forecasts total AOD (all aerosols at 550nm)
averaged over 1 – 28 August, 2017. The AOD peaks over the Western Sahara, near the coast of
West Africa, and in the Bodéléle Depression, within the interior of the continent, which are
consistent with source regions over summertime in North Africa (Engelstader and Washington,
2007). The AOD, however, overestimates the hotspots by ~25% when compared to the summer
AOD climatology from the Modern-Era Retrospective analysis for Research and applications
(MERRA, v2) (Randles et al. 2016). Nonetheless, the use of NGAC does not affect our
qualitative interpretation of the aerosol-affected radiances on the analyses and forecasts.

We also conducted short-range forecasts in each experiments' fully cycled system. To do

this, the forecast model within GFS runs 120-hr weather forecasts at T670 (~30km) resolution,
which are initialized on 00 UTC of each day (Fig. 1: green). The forecast model does account for
aerosol radiative effects using prescribed monthly aerosol climatologies from OPAC (Hess et al.
1998). But for both experiments, we use the *same* configuration in the forecast model, which
means that changes to the forecasts arise solely by the model's response to the analysis
differences, rather than the physics driving the forecast model.

To demonstrate the aerosol impact on the IR radiances, Fig. 3 shows a timeseries of each

experiment's observation-minus-forecast (OMF) BT for an IR channel (12.93 um) from the
Infrared Atmospheric Sounding Interferometer (IASI); the channel and sensor are representative
for other IR window channels and thermal IR sensors, respectively. For both experiments, Fig. 3
shows that the OMFs, which are averaged over North Africa and the East Atlantic, have a similar
root-mean-square (RMS) (top) and negative, or cold, bias (bottom) during the period of interest.
But for the cold bias, the AER run (red) is slightly more positive than the CTL run (blue). This
reduction in the cold bias for AER is due to the incorporation of aerosol transmittance effects on
the forecast (simulated) BT (via scattering). The average impacts are small (~1.7 K) over the
region, but the bias differences can be substantial (up to ~10 K) in localized regions during
strong Saharan dust events (Sokolik et al. 2001). When the aerosol-affected OMFs are
assimilated, this produces warmer analyzed temperatures at low-levels in the atmosphere
(Weaver et al. 2003; Kim et al. 2018; Wei et al. 2021).
*2.2 Wave tracking*

To identify the synoptic wave patterns during the period of interest, we used an objective

tracking algorithm similar to that in Brammer and Thorncroft (2015). Briefly, the tracking
algorithm involves analyzing mass-weighted centers of vorticity at multiple levels (i.e., curvature
vorticity at 850, 700, and 500 hPa; relative vorticity at 850 and 700 hPa). The wave center is then
determined from a weighted average of the centers within a specified radius (500 km). For each
experiment, the wave centers were extracted using the 6-hourly analysis fields, which identified
several systems that traversed North Africa and the East Atlantic. The tracking included waves
that later developed hurricanes, which we focus on in this study given their long lifetimes and
downstream implications.

For the time period of interest, two hurricanes developed from AEWs: Gert and Harvey.

Figure 4 shows the objective track locations for the AEWs that developed Hurricanes Gert and
Harvey in the CTL run over North Africa and the East Atlantic. For Gert (solid line), the storm
originates over Northeast Africa, at 5 – 10°N, on the 31$^{st}$ of July and moves northwestward over
North Africa before reaching the East Atlantic on the 4$^{th}$ of August. In contrast, Harvey (dotted
line) originates from two vortices over North Africa, at 25 – 29°N and 8 – 12°N, that develop on
the 8$^{th}$ of August and merge into one vortex near the coast, on the 12$^{th}$ of August; the storm then
moves west/southwest over the East Atlantic. Both waves developed hurricanes while over the
western portion of the Atlantic Ocean.

Comparison of the track locations for CTL and AER show little difference in the storm

positions during their evolution (not shown). After the initial development, the track locations
among the two cases are less than 250 km. Given the wavelength of the AEWs span 2000 – 5000
km (Burpee 1974), the aerosol-aware assimilation does not appear to have a significant influence
on the wave tracks. Therefore, we use track locations from CTL when investigating the storm
structures in the analyses and forecasts for both cases.
**3. Results**
*3.1 Analysis Differences: Time-average fields*

Before investigating the AEW cases shown in Fig. 4, we first examine the aerosol

impacts on the time-averaged background temperature, background zonal wind, and AEW
meridional wind variances.
Figure 5 shows cross-sections of the time-averaged background temperature and zonal
wind for CTL (contours) and the AER – CTL difference (colors) averaged over 1 - 28 August,
2017. Consider first the CTL run. The experiment captures the main summertime circulation
features over the region. For temperatures, the warmest air is positioned near the surface over the
Saharan Desert (Fig 5a: 20°N-30°N). This warming sets up a strong meridional temperature
gradient that extends vertically up to ~650 hPa and horizontally across the Sahel and over the
East Atlantic (Fig. 5b: 30°W-20°E). For the zonal wind, there is a well-defined AEJ at 650 hPa
(Fig. 5c: 15°N) that extends across North Africa and the East Atlantic (Fig. 5d: 20°W – 15°E,
10°N – 15°N) and low-level westerlies (800-1000 hPa) that are associated with the West African
Monsoon (WAM) flow (Fig 5c: 8°N-18°N).
The AER – CTL differences in Fig. 5 indicate how the aerosol-affected radiances impact
the time-averaged background fields. For temperature, the aerosol impacts warm the boundary
layer over the Sahara and Sahel by ~0.5 K (reddish colors in Fig. 5a: 10°N – 30°N, 1000 hPa –
650 hPa) and cool the marine boundary layer below the SAL by ~0.5 K (blueish colors in Fig.
5b: 15°N – 25°W, 15°N – 30°N). These temperature changes are qualitatively consistent with
enhanced aerosol heating in the boundary layer over the continent and in the SAL offshore. Over
land, the heating peaks at 800 hPa in the Sahel and the southern Saharan Desert (Fig 5a: 15°N -
25°N). The location of the heating indicates that the aerosol-aware assimilation: (i) increases
lapse rates (or reduces static stability) below the peak heating (1000 – 800 hPa) in the Sahel and
southern Sahara and (ii) enhances the meridional temperature gradient below the AEJ (1000-650
hPa) across the Sahel.
The AER – CTL differences in temperature support the changes to the background zonal
wind via adjustments to the thermal wind. For example, along the enhanced meridional
temperature gradient, AER accelerates the AEJ by ~0.5 m s$^{-1}$ (blueish colors in Fig. 5c: 10°N –
15°N, 700 – 600 hPa, and Fig. 5d: 20°E – 30°W, 10°N – 15°N), and accelerates the westerly
flow of the WAM by about ~1.0 m s$^{-1}$ (reddish colors in Fig. 5c: 12°N – 19°N, 1000 – 850 hPa).
Away from these features, the structural changes to the zonal wind are more difficult to interpret.
But inspection of the shear difference plots show that the aerosol-aware assimilation: (i)
increases the vertical shear below the AEJ (15°N – 22°N, 900 – 700 hPa) and (ii) decreases the
horizontal shear on the flanks of the AEJ axis (8°N – 18°N, 800 – 600 hPa) (not shown).
Figure 6 shows a vertical cross-section of the time-averaged, 2-6 day filtered meridional
wind variances, which is a proxy used to assess AEW amplitudes (Reed et al. 1988; Pytharilous
and Thorncroft 1999). The filtered meridional wind variances capture the two AEW tracks over
the interior of North Africa (contours show the CTL run). For both experiments, the wave
structures peak at levels consistent with AEWs examined in previous studies (south: 8°N – 13°N,
700 – 600 hPa; north: 18°N – 22°N, 950 – 800 hPa). But the AER – CTL differences (colors)
show that for the AER run, the meridional wind variances increase by ~15% in the northern
vortex and decrease by ~10% in the southern vortex. Note that the AER run also increases the
wind variances near the AEJ core by ~25% (15°N, 600 hPa), but this increase does not change
the peak location of the southern vortex.
The differences in the AEW meridional wind variances shown in Fig. 6 are, in part, due
to changes to the background fields, which can be explained by the local wave energetics
(Norquist et al. 1977; Hseih and Cook 2005; Bercos-Hickey et al. 2020). In absence of diabatic
processes, the AEW's southern structure extracts energy from the background via barotropic
conversions, which are proportional to the horizontal shear of the AEJ, while the northern
structure extracts energy via baroclinic energy conversions, which are inversely proportional to
the static stability (Thorncroft and Hoskins 1994; Paradis et al. 1995; Thorncroft 1995). This
means that for AER, the changes to the background zonal wind and temperature (i) reduce wind
variances in the southern vortex via decreased horizontal shear on the equatorward side of the
AEJ (barotropic) and (ii) increase wind variances in the northern vortex via reduced static
stability below the AEJ (baroclinic).

The qualitative explanation of how aerosol-affected radiances impact the waves via the

background fields aligns with the first of two pathways in which dust can affect AEWs
mentioned in the introduction. That is, the aerosol-aware assimilation captures dust radiative
effects that operate on the analyzed background temperature, AEJ, and thus the AEW wind
variances. But it is worth mentioning that dust radiative effects are also coupled to the forecast
model (i.e., from the OPAC aerosol climatology), which operate on the analysis fields via the
first-guess meteorological fields. Thus in AER, changes to the time-averaged fields in Figs. 5
and 6 are due to the NGAC aerosols in the assimilation system modifying existing radiative
effects imposed by the OPAC aerosol climatology in the forecast model.
*3.2 Analysis Differences: AEW cases*

In this subsection, we examine the impact of the aerosol-aware assimilation on the AEW

analysis fields for our cases described in Section 2.2.

Figure 7 compares the structure of the AEW that developed Gert for CTL and AER. The

AEW crosses Africa and the East Atlantic from 31 July – 4 August. During these times, the wave
remains south of the AEJ and is thus largely away from the dust aerosols. But despite this
separation, the aerosol-aware assimilation affects the evolution of the wave structure (Fig 7a, 7c:
colors surrounding the X's). For example, on the 2[nd] of August the AER run decreases the wave,
which at this stage is an open trough (Fig 7a: blueish colors surrounding the X). The vertical
structure also shows that the cyclonic vorticity for AER (red) is ~10% less than for CTL (blue)
from 600 – 800 hPa (Fig. 7b). On the 4[th] of August, the wave intensifies as it moves offshore,
forming a closed streamline circulation (Fig. 7c). But similar to the onshore wave, the aerosol
impacts on the vertical structures continue to reduce the vorticity within the storm center by
~10% (Fig. 7d).
Figure 8 compares the structure of the AEW that developed Harvey for CTL and AER.
The AEW develops as two vortices over East Africa on the 8[th] of August, and travels westward.
On the 9[th] of August, the land-based AEW is broad in structure and covers a large portion of the
continent (Fig. 8a). For AER, there are strong changes within both vortex centers, which include
increases in the vorticity around the northern vortex (reddish colors at 18°N) and decreases in the
southern vortex (blueish colors at 14°N). The vertical structures show that vorticity for the
northern vortex is, on average, ~20 – 35% larger from 600-850 hPa (Fig. 8b: cf. solid blue and
solid red), while the southern vortex is ~20 – 35% smaller from 750-850 hPa (Fig. 8b: cf. dotted
blue and dotted red). On the 12[th] of August, the two vortices merge into a single wave offshore.
Compared to the land-based AEW, the amplitudes of the combined wave are weak and its
vertical structure changes little with height (Fig 8c, 8d). Consequently, the aerosol impacts are
reduced, affecting the vorticity by ~5-15% from 1000-500 hPa (Fig. 8d).
Over Africa, the aerosol impacts on the AEWs for Gert and Harvey were consistent with
the time-averaged AEW meridional wind variances in Fig. 6, but the impacts were stronger for
Harvey. The story is different offshore as the impacts remain moderate for Gert but weaken for
Harvey; the latter may be due to the merging of the vortices and the positioning of the aerosols.
Therefore, we focus on the land-based stage of the AEWs and further investigate the aerosol
impacts.
To understand how the aerosol-aware assimilation impacts our AEW cases, it is
informative to examine the episodic dust plumes and radiance observations. Figure 9 shows a
snapshot of the NGAC AOD (brown contours) for times when the AEW for (a) Gert and (b)
Harvey are over Africa; the X's mark the position of the vortex centers. Overlaying the AOD are
observations from the IASI sensor at the same time; shown are the AER – CTL differences in the
BT at 12.93μm (circles), the same sensor and channel shown in Fig 3. For Gert, the BT
differences surrounding the wave are negative.  This indicates that near the wave center, the BTs
are cooler in the AER run (Fig. 9a), but the values are small (light blue circles). In contrast, for
Harvey, the negative values are largest near the northern vortex (dark blue circles), which is also
immersed in a dust plume with AODs over 1.0 (Fig. 9b).
When aerosol-affected radiances are assimilated, warmer analyzed temperatures are
typically produced at low-levels over North Africa and the East Atlantic (Kim et al. 2018; Wei et
al. 2021). For the AEW that developed Gert, the degree of warming over Africa is similar to the
time-averaged AER-CTL background temperatures shown in Figs. 5a and 5b. But for the AEW
that developed Harvey in AER, the temperatures over the wave's northern vortex (18-22°N)
warm as much as 1.5 K at mid-levels, 900-600 hPa, which is double the time-average. The
implications of this additional warming on the AEW vorticity is explained below.
Grogan and Thorncroft (2019) showed through energetic arguments that the heating from
an episodic dust signal that interacts with the AEW's northern vortex generates eddy available
potential energy (APE ~ $T'^2$). Previous idealized studies have also shown that dust-induced eddy
APE amplifies the northern structure of AEWs (Grogan et al. 2016, 2019; Nathan et al. 2017;
Bercos-Hickey et al. 2017). For the Harvey case in the AER run, the scenario is the same as in
Grogan and Thorncroft (2019), but the aerosol-affected radiances capture the heating from the
dust plume, rather than the forecast model, which in turn drives the amplified vorticity in the
AEW's northern vortex.
The impact of the episodic dust plume on the northern vortex for the AEW that
developed Harvey aligns with the second pathway in which dust can affect AEWs mentioned in
the introduction. Thus the combined effects of both pathways may help to explain why the
aerosol impacts for the AEW with Harvey are stronger than the AEW with Gert.
3.3 *Forecast Differences: AEW cases*
To examine the impact of the aerosol-aware assimilation on the forecasts for our AEW
cases, we compare the Root-Mean-Square-Error (RMSE) in vorticity for CTL and AER; the
forecasts were verified against their respective analysis. Table 1 shows the RMSE relative
differences between AER and CTL for the 1000 – 500 hPa vorticity following the AEWs. To
compute the RMSE following the AEW at each forecast time, we use the CTL wave locations
shown in Section 2. For Gert, a 10° latitude by 10° longitude window is centered on the wave.
For Harvey, our window over North Africa has a fixed latitude of 5 – 25°N and a 15° longitude
range that is centered on the two vortices; over the Atlantic Ocean, a 10° latitude by 10°
longitude window is centered on the merged vortex.
Table 1 shows the AER run produces neutral improvement in the forecasting of the AEW
that developed Gert, as evidenced by the mixture of red and green values in the RMSE relative
differences. Inspection of the forecasts show that both AER and CTL underestimate the
intensification of the AEW when initialized onshore, on 31 July – 2 August, and overestimate the
intensification when initialized offshore, on the 3[rd] of August. As a result, there were several
instances where the RMSE forecast differences did not produce statistically significant results
(i.e., crossed out values for Gert in Table 1).
In contrast to the AEW that developed Gert, Table 1 shows the AER run produces
statistically significant improvement in forecasting the AEW that developed Harvey. The largest
improvements are found for the forecasts initialized on the $10^{th}$ and $11^{th}$ of August, with the
forecast on the $10^{th}$ showing reductions in RMSE for every forecast day (errors reduced by ~15-
49%). For the initialized times that we examine for Harvey (8 – 11 August), both the analyzed
amplitudes and AER – CTL vorticity differences were larger than Gert while onshore (cf. Figs. 6
and 8). Inspection of the forecasts reveal that the CTL run continues to suppress the wave
amplitudes downstream, while the AER run better maintains the intensity of the wave as the two
vortices merge over the East Atlantic and travel downstream.
In summary, the forecast error of the 1000-500 hPa averaged vorticity for the AEW that
developed Gert are similar among the two experiments, but dramatically reduced in AER for the
AEW that developed Harvey. This marked improvement with Harvey is likely associated with
the aerosol-aware assimilation capturing radiative effects of the large-scale Saharan dust plume
that interacted with the AEWs northern vortex. Therefore, ingesting mixing ratios of episodic
aerosols to constrain radiance calculations within the assimilation system can improve
forecasting the evolution of AEWs.
**4. Conclusions and Discussion**
In this study, we examined how incorporating time-varying aerosols into the assimilation
of satellite radiances affected the analyses and forecasts from GFS v14 and the corresponding
GDAS. In particular, we investigated the impacts of Saharan dust on AEWs and their
environment over North Africa and the East Atlantic during August 2017. To do this, aerosol
forecasts from the NGAC, v2 model were ingested into GDAS and constrained to the radiance
calculations to produce analysis fields (aerosol-aware) that were compared to a control
experiment that excluded aerosols (aerosol-blind). The analysis fields from both cases were then
used to forecast two AEW cases during our time period that were structurally different over
Africa, but later developed Hurricanes Gert and Harvey over the Atlantic Ocean.

Analysis differences showed that the aerosol-aware assimilation affected several fields

over North Africa and the East Atlantic. For example, the aerosols warmed the Saharan boundary
layer, accelerated the AEJ and the westerlies associated with the WAM, and modified AEW
meridional variances, with amplitudes increasing within the northern vortex and decreasing in
the southern vortex. The changes in the AEW meridional variances were also consistent with the
vorticity changes for the individual AEW cases examined.

The impact of the analysis differences on forecasting our AEW cases depended on the

wave structure. For the AEW that developed Gert, which did not have a northern vortex, RMSE
differences showed that the aerosol-aware experiment produced neutral improvement to the
forecasts of the vorticity field tracking the wave over North Africa and the Atlantic. But for the
AEW that developed Harvey, which had a northern vortex, the aerosol-aware experiment
improved the vorticity field in most forecasts. Moreover, the largest reductions in RMSE
occurred when analysis differences in the AEW structures were largest.

In exploring the results, we showed qualitatively that the aerosol-aware experiment (via

NGAC aerosols) captured the two pathways involving dust radiative effects on the AEWs, i.e.,
through dust-induced changes to the AEJ and background temperature fields (first pathway), and
through the interaction between the episodic dust plumes and the waves (second pathway). For
example, the aerosol-aware experiment modified the analyzed background temperature and AEJ,
which in turn modified the analyzed time-averaged AEWs that is consistent with barotropic and
baroclinic instability. Additionally, the aerosol-aware assimilation captured the enhanced
warming and vorticity associated with the formation of an episodic dust plume interacting with
the northern vortex of the AEW that developed Harvey. The aerosol impact on the AEW that
developed Harvey is similar to dust-coupled AEWs shown in Grogan and Thorncroft (2019). In
contrast, the impact is absent in the AEW the developed Gert because the wave did not have a
northern vortex nor interact with a dust plume.
The improvement on forecasting the AEW that developed Harvey suggests the
importance of the aerosol-aware assimilation capturing dust radiative effects on AEWs involving
episodic dust plumes. Although the AEW that developed Gert was influenced by the aerosol
transmittance effects on the time-averaged background fields, this did not improve forecasting of
the storm. Therefore, investigating more cases that do and do not interact with episodic dust
plumes would better determine the utility of our approach for forecasting AEWs. Moreover,
there are known variabilities in AEW activity (Brammer and Thorncroft 2017) and dust source
regions over West Africa (Wagner et al. 2016), and therefore different scenarios of the AEW-
dust plume interaction should be examined.  Nonetheless, forecast improvements such as those
shown for the AEW that developed Harvey are encouraging and could be critical for determining
the timing and location of tropical cyclogenesis that originate from developing AEWs.
Aerosol radiative effects can be incorporated into the NWP system through the forecast
model and through the assimilation system. Though few studies focus on the assimilation
approach, such as Kim et al. (2018) and Wei et al. (2021), this study has demonstrated the
importance of incorporating time-varying, episodic aerosols into the satellite radiance
calculations to capture dust radiative effects on the analyzed AEWs. More work, however, is
needed to better understand how to optimize the aerosol-aware assimilation, such as adjusting the
bias-correction and quality-control procedures (Wei et al. 2021). Moreover, future work should
investigate how much complexity is needed to represent aerosol processes adequately and
accurately, and thus effectively account for aerosol effects within the NWP system.
**Data availability**
Analyses and forecasts from the AER and CTL runs can be provided upon request to the
first author of the paper.
**Author contributions**
DG and SL developed the ideas for the study. SW and SC conducted the numerical
experiments. DG, CL, and SW analyzed and interpreted the results. DG prepared the paper. DG,
CL and SW reviewed the paper.
**Competing interests**
The authors declare that they have no conflicts of interest.
**Acknowledgements**
The work presented here is supported by NOAA NWS NGGPS R2O (Award number
#NA15NWS468008). The NWS project is a collaborative effort from the University at Albany
(Cheng-Hsuan Lu, Shih-Wei Wei, Sheng-Po Chen, and Dustin Grogan), NCEP/EMC (Robert
Grumbine, Andrew Collard, Jun Wang, Partha Bhattacharjee, Bert Katz, Xu Li), and
NESDIS/STAR (Quanhua Liu, Zhu Tong). The GDAS experiments were conducted at the
University of Wisconsin-Madison Space Science and Engineering Center's Satellite Simulations
and Data Assimilation Studies computer, or S4, cluster.

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

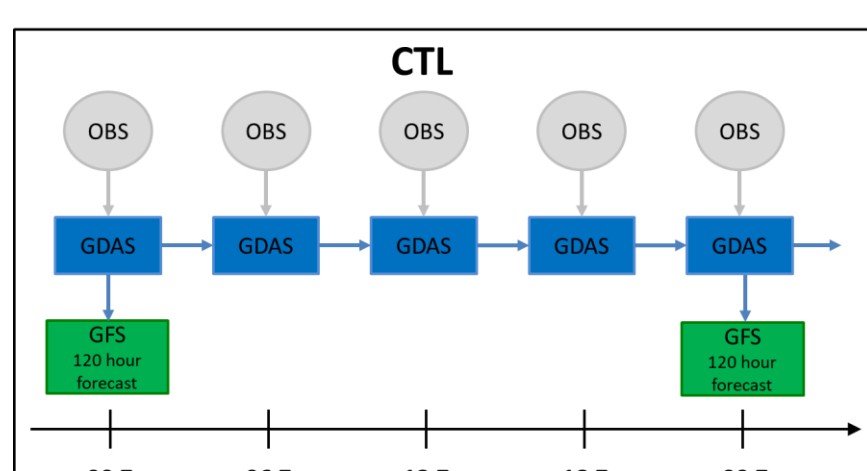

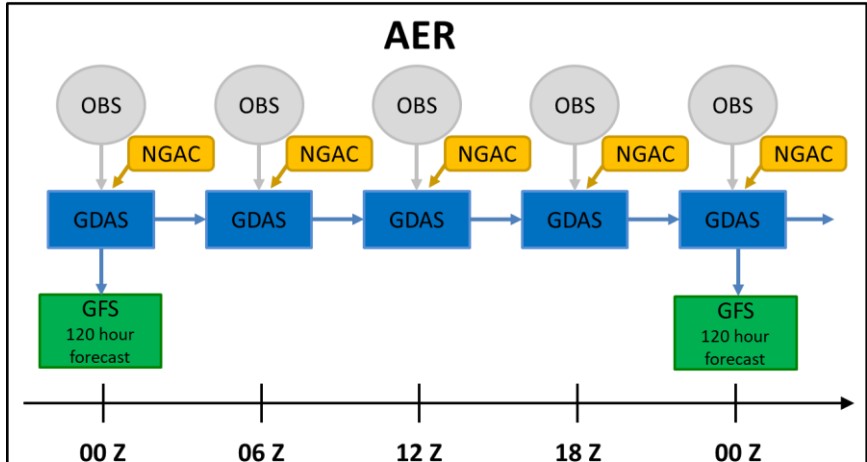



**Figure 1.** Schematic flow chart of the aerosol-blind (CTL) and aerosol-aware (AER) experiments in this study. See text in section 2.1 for details.


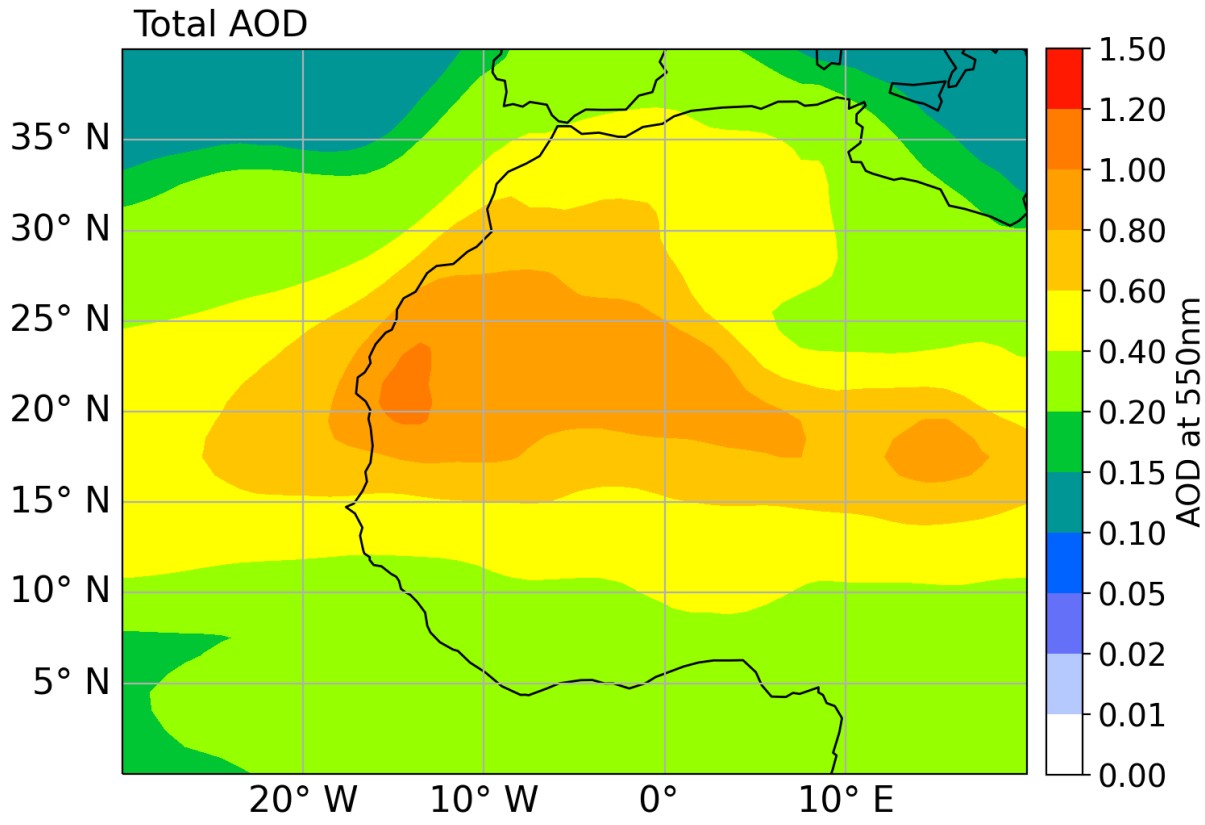


**Figure 2.** Total Aerosol Optical Depth (AOD) from the NGAC forecasts, averaged over 1-28 August, 2017.

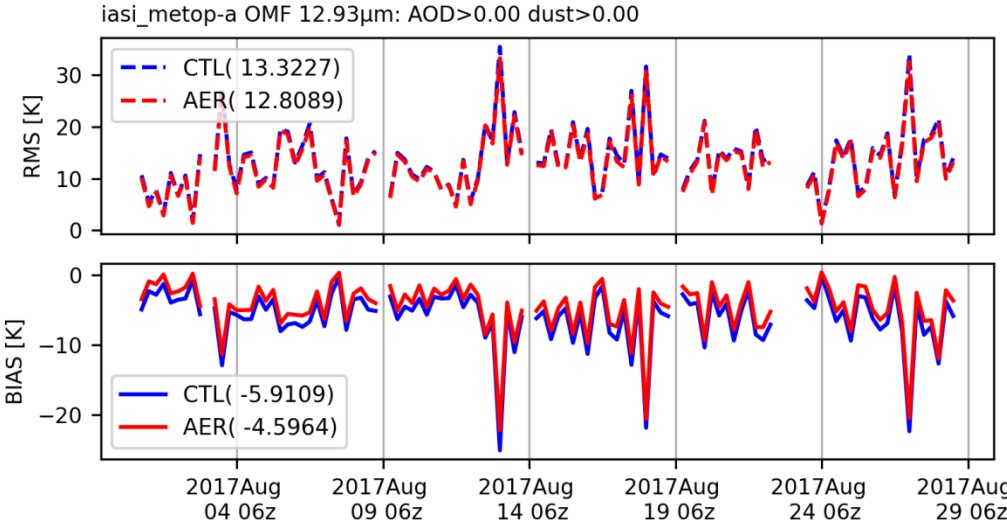


**Figure 3.** Statistics for the observation-minus-forecast (OMF) infrared brightness temperatures (IR BT) (12.93μm) from the IASI
hyperspectral sensor from CTL (red) and AER (blue). The timeseries includes all observations over the region (0-40°N, 20°E-
30°W), irrespective of aerosol loading. The numbers in the legend are the mean values for the (top) RMS and (bottom) bias for
each experiment.


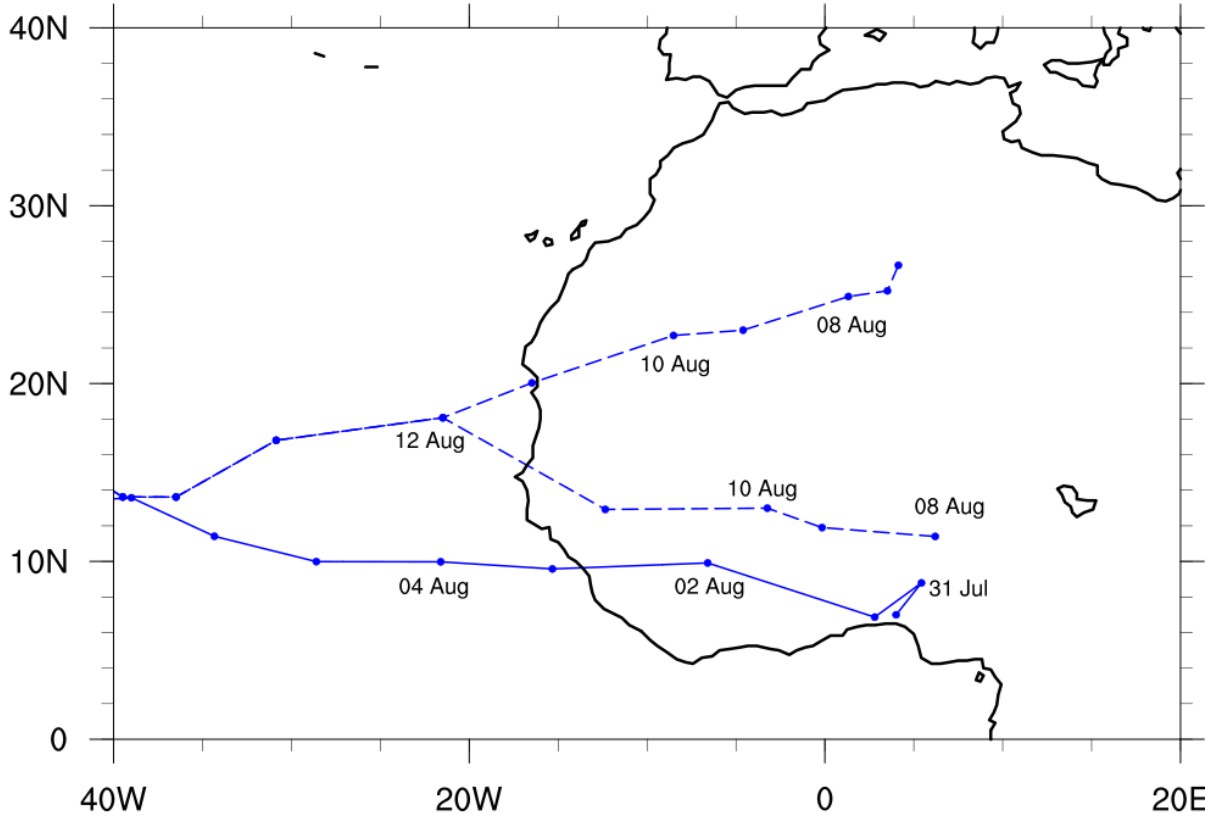


**Figure 4.** Daily locations (at 00 UTC) of the AEWs corresponding to Gert (solid) and Harvey (dashed) obtained by the tracking
algorithm in the CTL run (time period: August 2017).

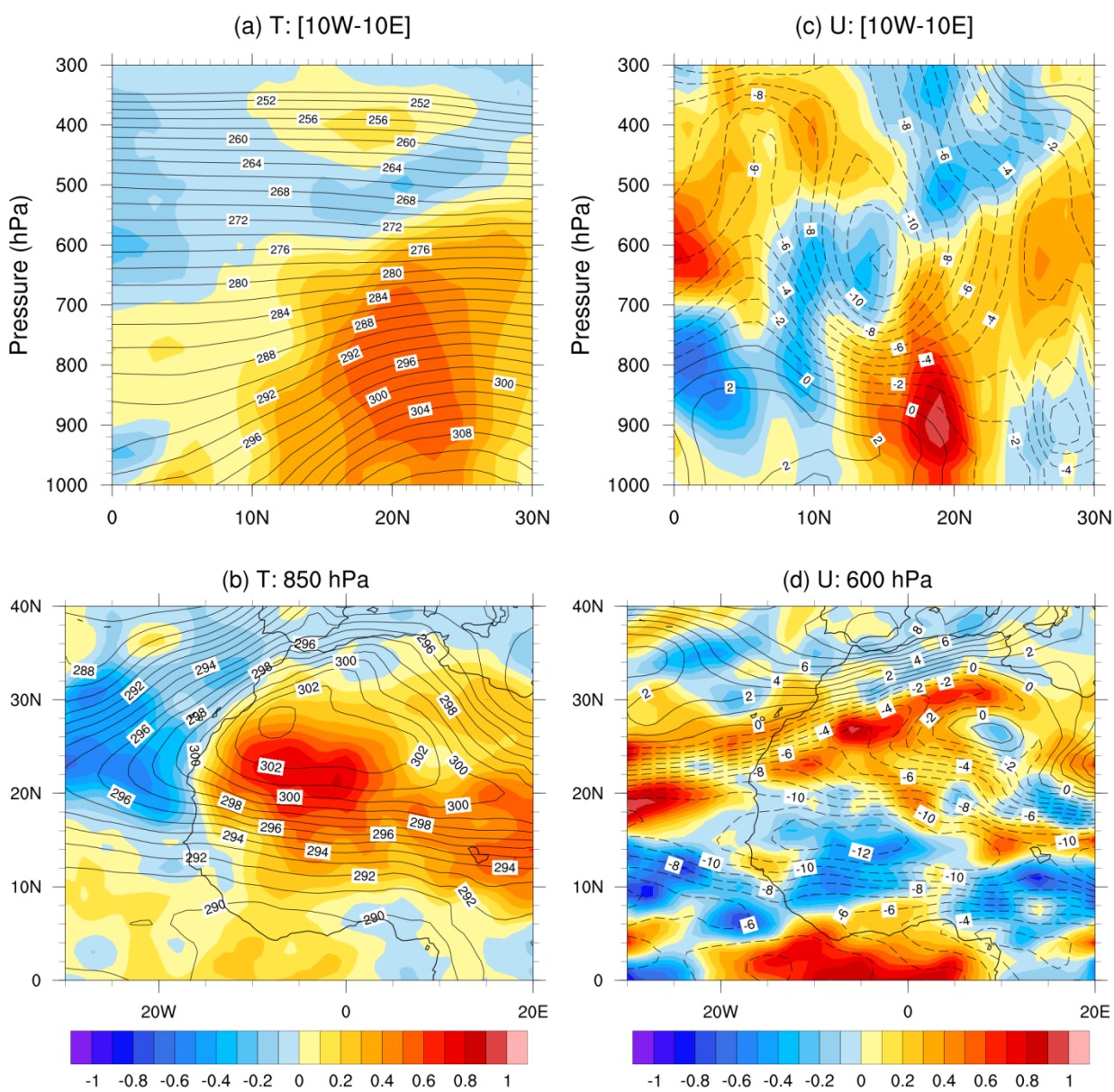

**Figure 5.** Vertical and horizontal cross sections of the CTL analysis (contours) and the AER – CTL analysis difference (colors) for (a, b) temperature, T, and (c, d) zonal wind, U. The vertical sections (top) are zonally-averaged from 10°W – 10°E, while horizontal sections (bottom) are taken at specified pressure levels. Contour/color units: (a,b) K and (c,d) ms$^{-1}$. The fields are time-averaged from 1 – 28 August, 2017.

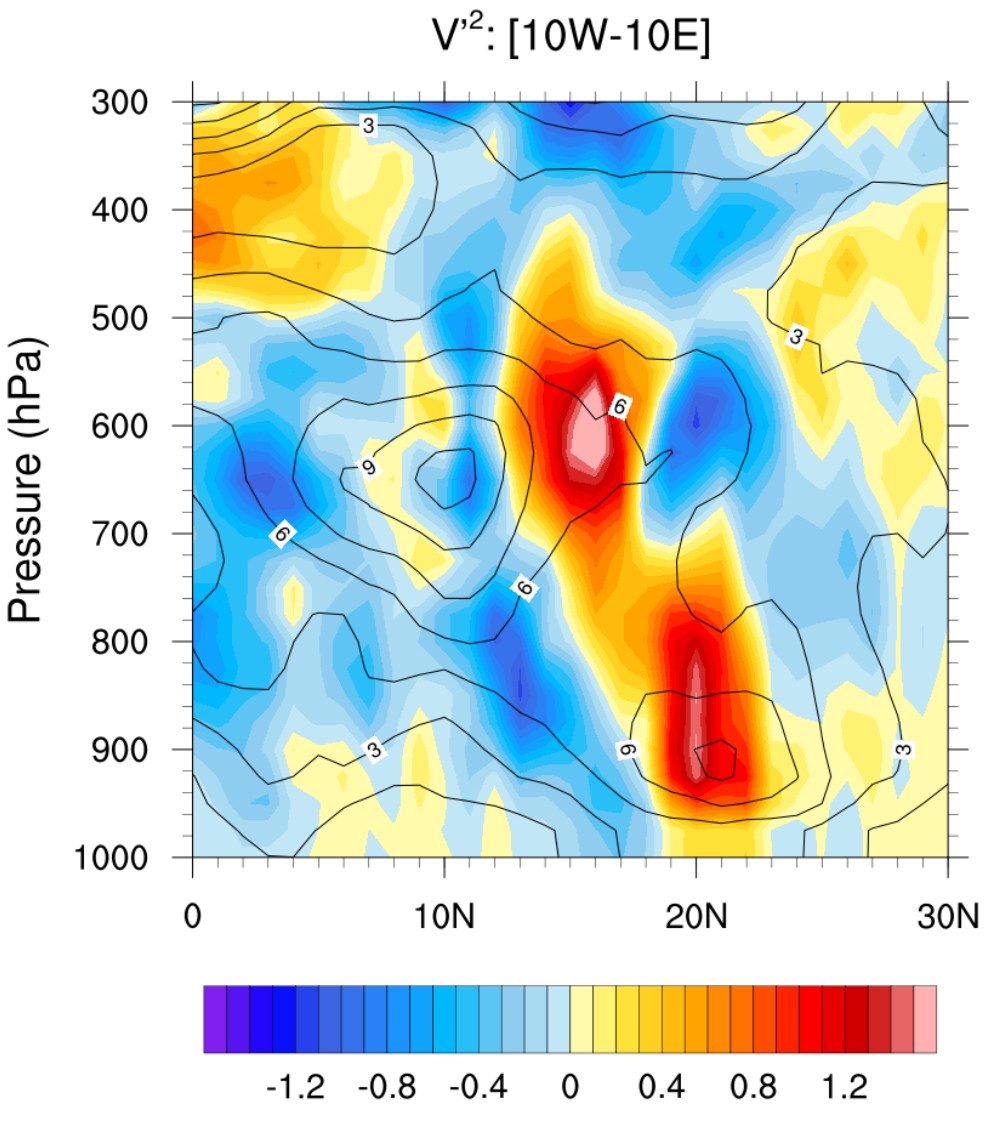


**Figure 6.** Time-averaged 2-6 day filtered meridional wind variances, v'$^2$, of the CTL analysis (contours) and the AER – CTL
analysis difference (colors) zonally-averaged from 10°W – 10°E for August 2017. Contour/color units: m$^2$s$^{-2}$.








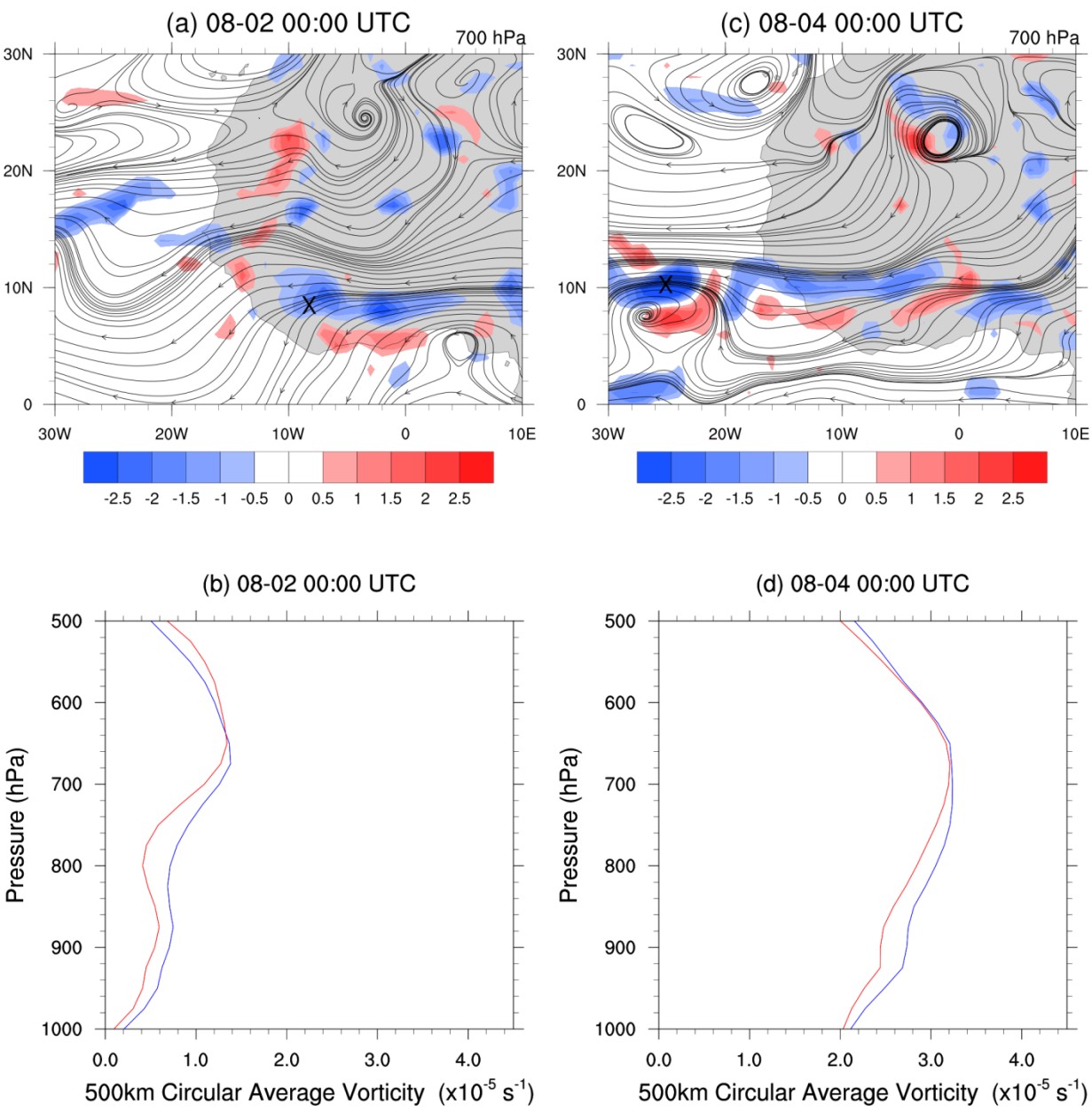


**Figure 7.** The evolution of the AEW associated with Gert on 2[nd] of August (left) and the 4[th] of August (right). The top panels show the 700 hPa CTL streamlines (black) and the AER – CTL 700 hPa cyclonic vorticity differences (red/blue); the 'X' marks the wave's location from the tracking algorithm. The bottom panels show the circular average vorticity (radius 500 km) taken at the X's for CTL (blue) and AER (red). Note that for the dates in the titles, the first digit corresponds to the month and the second digit to the day.


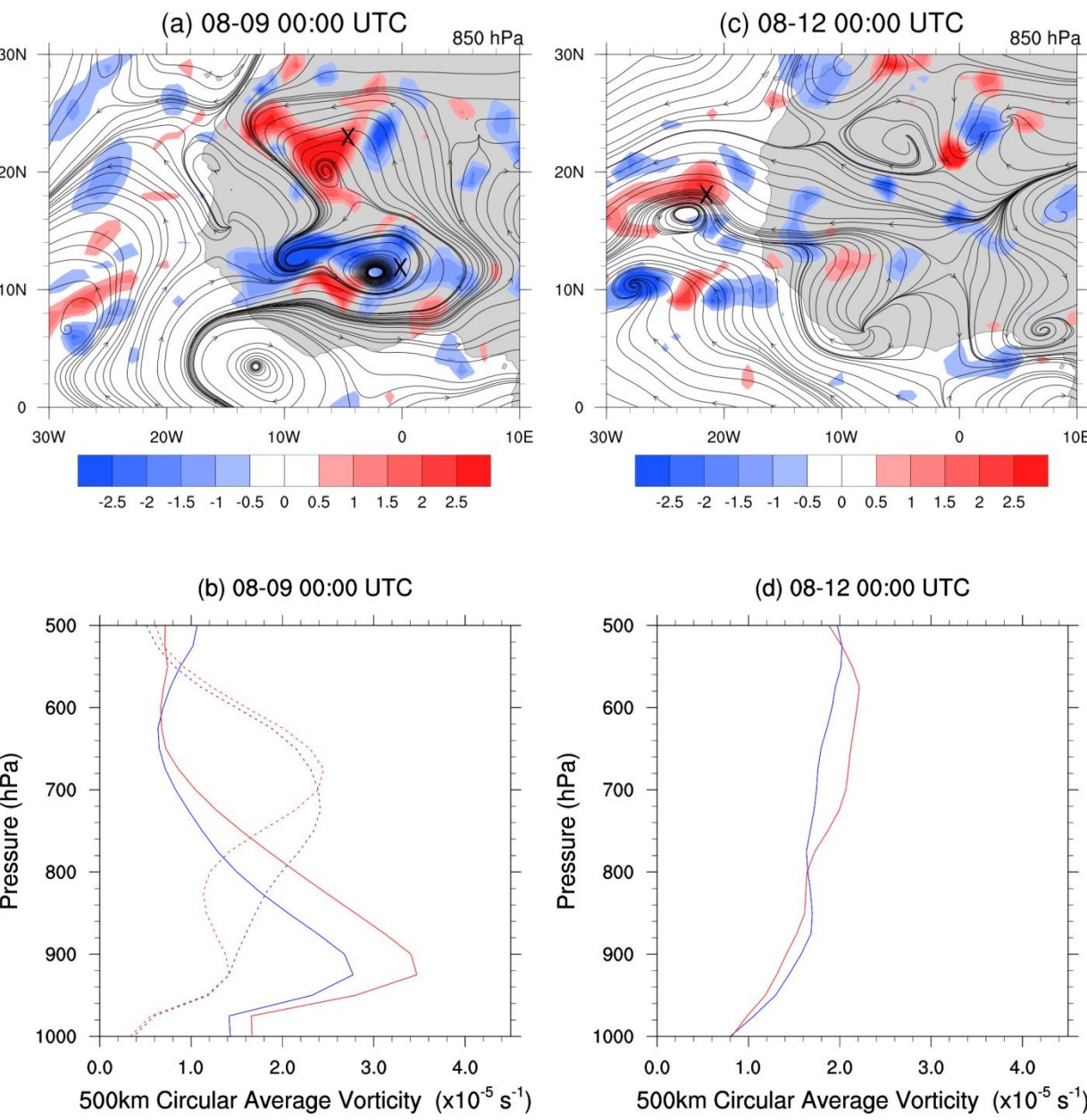

**Figure 8.** As in Fig. 7, but for the evolution of the AEW associated with Harvey on the 9[th] of August (left) and the 12[th] of August (right) The horizontal plots (top) show 850 hPa CTL streamlines and 850 hPa AER-CTL cyclonic vorticity differences, instead of 700 hPa, to better capture the two-vortex signal. Over Africa (b), we overlay the vertical vorticity structures of the northern (solid) and southern (dotted) vorticies for CTL (blue) and AER (red).

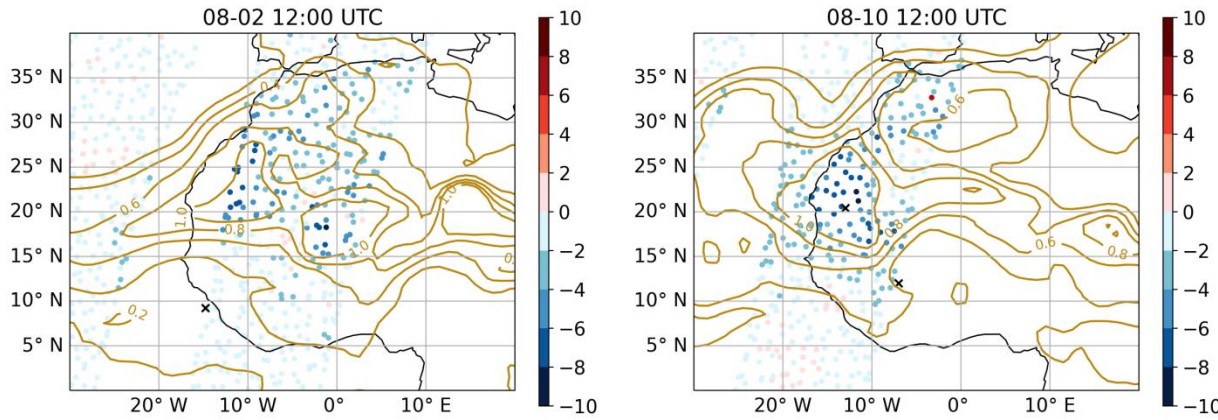


**Figure 9.** AER – CTL differences in simulated BT at 12.93μm from the IASI (colored circles) with the NGAC AOD (brown
contours) on the 2$^{nd}$ of August, 12:00 UTC (left) and the 10$^{th}$ of August, 12:00 UTC (right). The X's mark the location of the
wave centers for the AEW that developed Gert (left: 8°N,14°W) and Harvey (right: at 12°N,17°W and 20.5°N,13°W). Colorbar
units: K.









**Gert**

| Initialization | 1 day | 2 day | 3 day | 4 day | 5 day |
|---|---|---|---|---|---|
| 31 July | ~~0.13~~ | 0.21 | 0.19 | 0.38 | ~~0.03~~ |
| 1 August | 0.17 | 0.27 | 0.25 | ~~0.10~~ | ~~0.08~~ |
| 2 August | 0.19 | ~~0.04~~ | 0.24 | ~~0.10~~ | ~~0.08~~ |
| 3 August | ~~0.06~~ | 0.20 | 0.23 | ~~0.09~~ | 1.02 |

**Harvey**

| Initialization | 1 day | 2 day | 3 day | 4 day | 5 day |
|---|---|---|---|---|---|
| 8 August | 0.23 | ~~0.05~~ | 0.23 | 0.32 | 0.27 |
| 9 August | ~~0.08~~ | ~~0.07~~ | ~~0.06~~ | 0.33 | 0.32 |
| 10 August | 0.35 | 0.32 | 0.17 | 0.31 | 0.49 |
| 11 August | 0.22 | 0.39 | 0.49 | 0.46 | 0.64 |

**Table 1**. RMSE relative differences in the 1000 – 500 hPa relative vorticity between the AER and CTL forecasts for the AEWs
that developed Gert and Harvey. For each forecast day, the relative differences are calculated by taking (AER-CTL)/CTL of the
RMSEs over the region following the AEWs (see text for more details). The green values indicate AER improved the forecast,
while red values indicate AER degraded the forecast; crossed-out values were not significant to the 99% confidence interval. The
staircase border in each case separates times when the waves are located onshore (upper left) and offshore (lower right).