# Peer review of "Investigating the Impact of Saharan Dust Aerosols on Analyses and Forecasts of African"

_Atmospheric Chemistry and Physics, 2021_

## Author Response (AR1)

**Response to Reviewer 1 for**
**https://doi.org/10.5194/acp-2021-129-RC1, 2021**
**by D. Grogan, C.-H. Lu, S.-W. Wei, and S.-P. Chen**

We appreciate the Reviewer's careful reading of the manuscript. The Reviewer's suggestions, which we have fully addressed, have resulted in an improved manuscript.

**Summary:** *The assimilation of aerosol information is still not fully realised in most NWP systems and I thus welcome studies that explore potential impacts as this one. Overall I find the experimental design and outcome of this paper interesting and relevant for the readership of ACP, but the presentation of the results is not as clear and convincing as should be. The reference to relevant literature is a little thin, many figures are only superficially described and some important background information is missing. I therefore propose that major revisions are needed before I can recommend this work for final publication in ACP. Details are given below.* **Response:** We agree with the reviewer's assessment of the paper and thus have followed their guidance below, which has produced a much improved manuscript.

**Major Comments:**

**Title:** *I am not sure that this is an optimal title for this work. The second part read suite complicated and technical and I doubt people will really understand what that means. The first bit is fine but you leave out the aspect of TC genesis. You mention data assimilation but not forecasting. How about the following: Effects of Saharan dust assimilation on the analyses and forecasts of African easterly waves and Atlantic tropical cyclogenesis?* **Response:** We have changed the title of the manuscript to: "Investigating the Impact of Saharan Dust Aerosols on Analyses and Forecasts of African Easterly Waves by Constraining Aerosol Effects in Radiance Data Assimilation**.**" The revised title takes elements of the suggested title by the reviewer, but excludes dust data assimilation, as this study ingests dust for the sole purpose of incorporating it into radiance calculations.

**Abstract:** *The first paragraph reads well but the second is mostly a repetition of the first and should be omitted. Instead you should much better explain your experimental design (take information from aerosol model NGAC for the assimilation, change of initial condition but not of forecast model, comparison to satellite data). I would also explain that your results are consistent with qualitative arguments of baroclinic and barotropic instability.* **Response:** As suggested, the revised abstract has been combined into a single paragraph, includes a description of our experimental design, describes the time-averaged and case results, and connects the results to dust radiative effects operating on the background and episodic dust fields.

**Literature:** *Your introduction is relatively thin on references to relevant work. I would ask you to do a more thorough research and include more references. A few that come to my mind include Benedetti et al. (2018, ACP, overview paper for aerosol prediction), Knippertz and Todd (2010, JGR, dust and AEWs), Jones et al. (2003, JCLI, AEW and dust transport), Karyampudi et al. (1999, BAMS, Saharan air layer), Pope et al. (2016, GRL, dust data assimilation), Schwendike & Jones (2010, QJ, AEW merging and TCcyclogenesis),* **Response:** We have included most of the above mentioned references in the introduction, and several others throughout the manuscript. In total, 30 additional references have been added to the revised manuscript.

**Methodology:** *Your dust information come from NGAC and you only mention this in passing. I see that you are giving references here but this information is so essential for understanding the present paper that you need to give a summary here. How does this model work? What data is assimilated? How good is it? You need to justify your approach much better! In addition the radiative impact will depend a lot on the chosen optical properties. What is used here? How sensitive are the results to this? In the infrared the sensitivities can be quite large, as shown in a recent paper by Claudia di Bagio and co-authors!*

**Response:** The methodology has been bolstered significantly and addresses these questions. In particular, we now (i) provide more details of the experiments for each step of the workflow (i.e., gdas, obs, ngac and gfs), (ii) assess the NGAC aerosols, and (iii) examine statistics for the observed-forecast infrared brightness temperatures from IASI for each experiment.

**Introduction:** *In addition to more literature, your introduction should elaborate a clear hypothesis to test. There is a lot of literature on the link dust-radiation-temperature wind-AEWs-TCs including mechanisms of barotropic and baroclinic instability. The way the paper is written now, this link comes really late. I would bring this to the very front and cast it as a hypothesis you are testing. This would make the paper much more interesting to read and easier to understand. Doing this, you could even hypothesize that only the forecast of TCs that form from northern vorticies will be strongly affected by including dust, as the southern ones are too far away from the plumes. Harvey is a nice example, where the contribution of the northern vortex "wins"! You should stress this aspect more!* **Response:** We have refined the intro in the revised manuscript. The overlaying flow remains similar, but we have streamlined the front end to get to the motivation quicker. The literature involving dust-radiation-AEWs is retained and is drawn upon more in the paper for interpreting the results, making it more essential. For this study, our focus is to determine if dust radiative effects on AEWs identified in the literature can be captured within our aerosol-aware assimilation. Given that the study focuses on two cases, more cases may be needed to test if TCs that form from northern vortices will always be strongly affected by dust.

**Balance between sections 3 and 4:** *The description of results in section 3 is in places too short and superficial. Some aspects are then picked up again in section 4. I would merge these two for a better storyline. Make sure each panel of each plot is discussed in the paper. If not discussed, it can be omitted.* **Response:** In the revised manuscript, we have combined sections 3 and 4, removed unnecessary figures, and discussed all the panels within each figure.

**Conclusions:** *If you follow my advice on formulating a clear hypothesis, you can use the conclusion section to explain to what extent you find that hypothesis confirmed. You should also clearly articulate what we have learned from this study we didn't know before or in other words what is the innovation? Compare your results to those of other studies! Remind the reader of your very special methodology of taking dust from a model to assimilate into another model to change initial conditions to then make forecasts using a dust climatology. This is not straightforward and limits the interpretation of the results.* **Response:** The conclusions have been expanded in the revised manuscript. In the new version of the conclusions, we remind the reader of our methodology, present what we've learned in the study, and discuss the implications of the study on forecasting AEWs in NWP.

**Minor Comments:**

**Grammer Comments:** All minor comments pertaining to grammar have been fixed in the revision.

**General:** *Your geographical descriptions are often imprecise. What you call North Africa, I would call West Africa in some cases. What you call the southern Sahara, I would call the Sahel etc. Please check throughout and try to be consistent with common terminology.* **Response:** Fixed.

**Abbreviations:** *Make sure you define all at first use and then use abbreviation only. Don't define them, if not used again.* **Response:** We elect to redefine some acronyms in the methodology so that the section is self-contained.

**L200:** *This is confusing. The monsoon flow is southwesterly. You are showing that the westerly component accelerates but if the southerly decelerated, the total wind would not change. Please be clear or analyse both components.* **Response:** We now say on L264-L265, "the aerosols…accelerate… the

westerly flow of the WAM..." In the AER run, the southerlies of the WAM were also accelerated, but we exclude this because it has little relevance to AEWs.

**L201:** *Again confusing. I would call a positive change in an easterly flow deceleration not acceleration?!?* **Response:** The latitude range was incorrect in the original manuscript. We have removed this sentence in the revised manuscript to improve clarity.

**P8:** *I would explain the temperature first, then wind, as the latter is the consequence of the former.* **Response:** Fixed.

**L221:** *why modulus??* **Response:** We have removed the vorticity modulus figure and replaced it with the 2-6 day filtered meridional variances, which is a well-established proxy for AEWs. The results are the same.

**P12:** *The IASI data should be introduced in the method section. What exactly you do with them, is somewhat unclear to me. You need to describe this much better and discuss the results more clearly. Maybe good to show Gert, too, for contrast?!* **Response:** IASI is now mentioned in the methodology section. We also present statistics on the IASI infrared observations during assimilation to demonstrate differences among the two experiments in the methodology section. In the results section, Fig. 9 now includes one panel for Gert and one panel for Harvey, as suggested, to ease comparison and thus provide a better explanation for the results from this Figure.

**L313:** *better "the authors" to avoid repetition.* **Response:** The sentence referenced has been removed in the revised manuscript. The discussion pertaining to this sentence, which has been moved to section 3.2, is re-written to provide better clarity in explaining the results presented in Fig. 9. (L347-L354).

**Caption Fig. 1:** *Too short, more details here or reference to text*. **Response:** The caption refers the reader to the text for more details. Fig. 1 is now referenced throughout the methodology section of the revise manuscript

**Figs 2,3,4,5,9:** All suggestions have been adopted into the Figures.

We thank the reviewer for his/her very careful reading of the manuscript and for the suggestions for improving it.

**General Comments:** *The authors perform experiments with the GFS model which includes the radiative effects of aerosoles through data assimilation (GDAS). Those runs are called aerosol-aware runs. The control runs do not include the aerosol effect. Both types of runs were performed for the whole of August and two AEWs in 2017 which resulted in Hurricanes Gert and Harvey. The authors found for the time averaged analysis over August 2017, that in the aerosol aware run the AEJ and the WAM were accelerated and the temperature in the Saharan boundary layer increased, which lead to a modification of the vorticity structure and an increase in the northern and a decrease in the southern circulation. The authors also showed that in the aerosol-aware runs the errors of forecasting the AEW out of which Hurricane Harvey formed were reduced, but no improvements were found for the AEW out of which Gert formed. The paper is very well written, the aim of the paper and the results are very clear. I think this paper will be of interest to the scientific community. I only have very minorcomments.*

**Minor Comments:**

**Grammer Comments:** All minor comments pertaining to grammar have been fixed in the revision.

**p. 5, l. 137:** *Did both storms occur in this period? Seems like this is a period for Harvey and not Gert. Maybe this is coming later but it would be good to state somewhere for which period both sets of runs where computed.* **Response:** Yes, both storms occurred during our period of interest. In Fig. 2 of the revised manuscript, the tracks and the dates of the storms now overlay the figure.

**p. 7, l. 187-190:** *Are those averages for the whole of August and based on the 34 forecast runs you mentioned earlier? So far you only spoke about the period 25-28 July 2017. Better to say which data set those averages are based on.* **Response:** The time-averages are for August, which is now mentioned in the body and figure captions of the revised manuscript. We have also expanded the methodology to improve clarity on the workflow for the experiments.

**p. 9, l. 220:** *The text says "modulus" and the caption of Fig. 4 says "moduli". Why do you change between singular and plural? What exactly is a "relative vorticity amplitude modulus"? The caption says sqrt (zeta^2) is shown.* **Response:** In the revised manuscript, we have removed the vorticity modulus (moduli) figure and replaced it with the 2-6 day filtered meridional variances, which is a well-established proxy for AEWs. The results are the same.

**p. 9, l. 233:** *Have you averaged over 700 and 850 hPa to get the streamlines shown? The caption says only streamlines at 700 hPa are shown.* **Response:** No, the streamline and relative vorticity for the horizontal cross-sections occur at 700 hPa for Gert (Rev. Fig. 7) and 850 hPa for Harvey (Rev. Fig. 8). The change in elevation for Harvey is to better capture the two-circulation signal, which is now stated in the figure caption of Fig.8.
.
**Fig. 2:** *You could add the times that are shown in this figure to the caption. What are the dots referring to? 6hly times?* **Response:** The revised Figure 2 now has both storms on one plot and dates are shown for the storms while over West Africa.

**Fig. 9:** *Which unit is shown on the colour bar?* **Reponse:** the colorbar refers to the colors in the circles corresponding to the BT differences. This is clarified in figure caption of the revised manuscript.

We appreciate the reviewer's suggestions for improving the manuscript. Our responses follow.

**General Comments:** *Direct aerosol-affected radiance calculations are not practically adopted in current operational numerical weather prediction (NWP) and data assimilation (DA) systems. This is mainly due to computational cost issues. Also, uncertainties of land surface conditions in radiative transfer models contribute to the limitation. Thus, this paper are appropriate to the NWP development direction and requirement. The topics of the paper addresses the impact of aerosol-aware daiance calculation on the dynamical atmospheric structure on northern Africa. However, general recommendation is a major revision to the paper and additional experiments and evidences to draw a concrete conclusion and discussion. 2017 August time period was used in the experiment to investigate the dust impact on circulation patterns involving two Hurricanes cases, Gert and Harvey. The authors were able to identify that the aerosol-aware run reduces the errors of forecasting the African easterly waves. The improvement is positive especially for Hurricane Harvey case but neutral or no improvement for the Hurricane Gert case. Obviously additional experiment for different time period is needed for robust conclusion. General editorial comments about the current version of the paper: overall writing quality is not clear and additional literature survey is needed. Details are missing in figure and table captions and titles. At this stage, my opinion is to suggest major revisions and additional experiments for the paper. Detail editorial corrections and comments can be provided once a mature version is resubmitted. Nonetheless, a few early remarks and suggestions are given below.*
**Response:** In the revised manuscript, we have made the following changes to address the reviewer's concerns:
1. Refine the intro by including over 20 additional references, adding more motivation and a clearer hypothesis.
2. Bolster the methodology section by providing more details of the experiments for each step of their workflow (i.e., gdas, obs, ngac, and gfs), assessing the NGAC aerosols, and examining DA statistics for the infrared (IR) brightness temperatures from IASI for each experiment.
3. Combine sections 3 and 4 to streamline results with their explanation and remove unnecessary figures.
4. Expand the conclusions section to remind the reader of the methodology, present what has been learned in the study, and discuss the implications.
In regards to conducting additional experiments, we argue that the two cases, Harvey and Gert, are sufficient for this study. In particular, this study incorporates aerosol transmittance effects on satellite radiance calculations during data assimilation to (i) investigate their impact on the analysis and forecasts and (ii) explain the differences in the context of physical mechanisms driving dust radiative effects on AEWs. This study recognizes that more than one mechanism involving dust radiative effects is at play for the analysis fields of our two AEW cases, which we suggest is the reason for the improved forecast of Harvey and not Gert. Thus our study exposes the utility of our approach on AEWs interacting with dust. Nonetheless, we agree with the need for additional cases to increase the robustness of our results, which is touched on in the conclusion section of the revised manuscript.

**Main Comments:**

**1.** *Title slightly misleads discussion points. Is the main point about the effect of Saharan Dust on AEW from AGCM dynamics point of view or impact from DA procedures? Detail dust structures and distributions are not provided in the paper. DA and analysis statistics are not fully provided.* **Response:** We have changed the title of the manuscript to: "Investigating the Impact of Saharan Dust Aerosols on Analyses and Forecasts of African Easterly Waves by Constraining Aerosol Effects in Radiance Data Assimilation." The main point is to incorporate aerosol transmittance effects on satellite radiance

calculations to determine how, and to what extent, the assimilation captures dust radiative effects that operate on AEWs in the analysis fields, and what impact this has on forecasts for the AEWs downstream.

**2.** *Model experiments: Current operational version of the NCEP GFS system is based on the cubed sphere FV3 dynamical core and version number has already reached around version 16. GFS v14 used in the paper is considerably outdated. Prescribed monthly aerosol climatologies obtained from the OPAC package were applied in the experiments. It is very difficult to make any opinion about how useful the OPAC aerosol data sets are for direct applications in the NWP DA systems. Clearly, a trouble is to understand about the experiment design and approach: monthly climatological aerosol data set for one month NWP forecast and DA experiments.* **Response:** The revised manuscript presents the design and approach of our study more clearly and provides an in-depth description of the experiments conducted. Moreover, discussions involving the interplay between OPAC, from the forecast model, and NGAC, from the assimilation system, on the analysis fields are discussed in the context of dust radiative effects on AEWs.

**3.** *Figure 1 shows that NGAC data is used in the GDAS cycles. Again, DA analysis statistics of the aerosol-aware experiments are critically important for discussion.* **Response:** In the revised methodology section, we provide statistics for the IR brightness temperatures from IASI for each experiment.

**4.** *In the paper, mean forecast field differences are extensively compared for the experiments with and without aerosol-aware data assimilation. Since the experiments are based on the whole month of August, distinguishing aerosol background structures are key factors and following impact on the brightness temperature calculation should be provided for all assimilated infrared observation data sets. Single IASI scatter plot figure (in figure 9) is not sufficient.* **Response:** We now mention in the methodology that IASI, and other thermal IR sensor observations, are ingested in the assimilation system. As mentioned in comment 3 above, statistics are now performed for IASI on a channel in the IR window, which serves as a representative case for other IR channels and thermal IR sensors. In the revised manuscript, the single IR scatter plot now includes an example for Gert, which allows for clear comparison to explain the impacts on Harvey.

**5.** *Forecast RMSE differences are compared in Table 1 to identify the improvement. Obviously, there is a statistical risk to draw any conclusion with limited forecast samples.* **Response:** In the conclusions of the revised manuscript, we identify that additional work is needed to improve the robustness of our results, which includes the investigation of additional cases.

---

## Author Response (AR2)

**Response to Reviewers, Round 2**
**Investigating the Impact of Saharan Dust Aerosols on Analyses and Forecasts of African Easterly Waves by Constraining Aerosol Effects in Radiance Data Assimilation**
**by D. Grogan, C.-H. Lu, S.-W. Wei, and S.-P. Chen**

**Reviewer 1:** We appreciate the Reviewer's careful reading of the manuscript.

**Rev. 1 Summary:** *The revisions have improved the manuscript considerably. I'm happy for the paper to be published after the following very minor comments have been addressed.*

**Rev. 1 Grammar Comments:** All minor comments pertaining to grammar/language have been fixed in the revision.

**Rev. 1. Comment 1:** *Dates: I would avoid using country specific date formats and use, for example, 1-28 August 2017 instead.* **Response:** We have changed the country specific date formats throughout the revised document.

**Rev. 1 Comment 2:** *Fig. 3: It is slightly confusing that the zero lines are slightly offset. Are the tick marks of set as well? The last sentence in the captions says: "mean statistics". This is a bite vague. Could you be more precise?* **Response:** In the revised figure, we have added separation between the plots to better distinguish them from each other. We also now mention in the figure caption, "The numbers in the legend are the mean values for the (top) RMS and (bottom) bias for each experiment."

**Reviewer 2:** We appreciate the Reviewer's careful reading of the manuscript.

**Rev. 2 Summary:** *The authors did a good job in revising the paper and all important points are now addressed. However, there are still a few technical corrections needed before this paper can be published in ACP. Details are given below.*

**Rev. 2 Grammar Comments:** All minor comments pertaining to grammar/language have been fixed in the revision.

**Rev. 2 Comment 1:** *L48: why not give the exact dates here?* **Response:** We included the year of the storm to distinguish the hurricanes from other named storms. But because the time period is mentioned above, we have removed the year declarations (i.e., "Harvey (2017)" now simply reads "Harvey"). The exact dates of the storms are excluded from the abstract because it is not critical information. They are, however, included within the body of the manuscript.